# Automatic Cardiopulmonary Endurance Assessment: A Machine Learning Approach Based on GA-XGBOOST

**DOI:** 10.3390/diagnostics12102538

**Published:** 2022-10-19

**Authors:** Jia Deng, Yan Fu, Qi Liu, Le Chang, Haibo Li, Shenglin Liu

**Affiliations:** 1School of Mechanical Science & Technology, Huazhong University of Science and Technology, Wuhan 430074, China; 2York Region Secondary Virtual School, York Region, Markham, ON L3R 3Y3, Canada; 3Shenzhen Rehabilitation & Aiding Devices Industry Association, Shenzhen 518000, China; 4Wuhan Union Hospital, Wuhan 430022, China

**Keywords:** CPET, XG-BOOST, cardiopulmonary assessment

## Abstract

Objective: Among various assessment paradigms, the cardiopulmonary exercise test (CPET) provides rich evidence as part of the cardiopulmonary endurance (CPE) assessment. However, methods and strategies for interpreting CPET results are not in agreement. The purpose of this study is to validate the possibility of using machine learning to evaluate CPET data for automatically classifying the CPE level of workers in high-latitude areas. Methods: A total of 120 eligible workers were selected for this cardiopulmonary exercise experiment, and the physiological data and completion of the experiment were recorded in the simulated high-latitude workplace, within which 84 sets of data were used for XGBOOST model training and36 were used for the model validation. The model performance was compared with Support Vector Machine and Random Forest. Furthermore, hyperparameter optimization was applied to the XGBOOST model by using a genetic algorithm. Results: The model was verified by the method of tenfold cross validation; the correct rate was 0.861, with a Micro-F1 Score of 0.864. Compared with RF and SVM, all data achieved a better performance. Conclusion: With a relatively small number of training samples, the GA-XGBOOST model fits well with the training set data, which can effectively evaluate the CPE level of subjects, and is expected to provide automatic CPE evaluation for selecting, training, and protecting the working population in plateau areas.

## 1. Introduction

In high-altitude areas, people’s physical condition, training, and skills should be taken into special consideration [1]. Cardiopulmonary Endurance (CPE) is a core component of evaluation criteria and an important index in considering job placement, training and occupational risk in high-latitude workplaces [2]. Cardio Pulmonary Exercise Testing (CPET) is a cardiological test that measures the heart’s ability to respond to external stress in a controlled clinical environment [3]. Considering the paradigmatic changes in research, CPET provides more accurate and extensive experimental data, compared with traditional research methods such as walking or running tests. CPET also improves the accuracy of subsequent data processing and analysis. It has become an important clinical tool for assessing exercise capacity and predicting a prognosis in patients with heart failure and other cardiac diseases [4]. CPET provides a wealth of data for CPE evaluation. The most widely used parameters are maximal oxygen uptake (VO2max) and anaerobic threshold (AT) oxygen uptake. VO2max is the maximum oxygen uptake of the human body, which defines the maximum energy that can be obtained through aerobic metabolism during the maximum exercise period [4]. Studies have shown that VO2max can reflect the maximum exercise limit of the cardiopulmonary system and is a key predictor of cardiopulmonary function [5,6,7]. The anaerobic threshold (AT) is defined as the highest level of sustained activity at which the measured oxygen uptake accounts for the entire energy requirement [8]. AT has been shown to have a high correlation with exercise endurance, and has numerous clinical applications, such as prediction [9,10,11,12]. Exercise economy is used to define the energy consumption under a certain absolute exercise intensity, and VO2 represents the energy consumption of a certain physical labor or power output [13]. Carriere et al. showed that the identification of AT and RCP has a potential role in prognostic stratification of HFrEF [14]. Obviously, different parameters reflect various features of CPE, which are hard to be replaceable for each other; thus, it is widely accepted that multi-parameters should be collected for a more accurate cardiopulmonary endurance assessment. Traditionally, the multi-parameter interpretation has, nevertheless, placed more pressure on the expertise of professionals to interpret the data and draw a conclusive assessment [15]. Stringer used VO2max, ventilation, work rate and a series of data to study the cardiopulmonary function in patients with COPD [16]. Wagner et al. used VO2max, V-VCO slope and EOV to analyze and manage heart failure with reduced ejection fraction [17]. Remington et al. collected variables including VO2max, VE/VCO2 slope, VTand O2 pulse to study aortic stiffness [18]. These CPET interpretations are based on individual experts’ subjective analysis, which varied the results across subjects and testing scenarios. Moreover, clinical studies intended to set up universally applicable guidelines; however, the systematic review conducted by the Canadian Cardiovascular Society Guidelines identified that those guidelines are of “low” quality due to various factors such as small sample size, single-center design, heterogeneous patient populations, variable CPET methodology and outcome measurements [19]. This may lead to the underuse of CPET, despite its advantages in providing rich, useful information with noninvasive test procedures.

Researchers have attempted to apply machine learning to interpret CPET data, achieving suggestive results. Leopold et al. developed a greedy heuristic algorithm based on feature clustering to study the ability of CPET to predict the anaerobic mechanical power outputs [20]. Braccioni et al. used a random forest algorithm to analyze the relationship between symptoms and cardiopulmonary parameters of lung transplant recipients; this was based on incremental CPET [21]. Inbar et al. combined SVM with CPET data and obtained a prediction model with quite an accurate overall prediction ability [22]. Li et al. proposed a sequential clustering method to identify elite athletes based on CPET data including oxygen consumption, carbon dioxide, heart rate, and stroke output, but without good performance [23]. Gratiela et al. proposed a fuzzy evaluation method based on multiple physiological parameters such as VO2max, SV and HR to evaluate CPE levels of athletes and non-athletes with an accuracy of about 90% [24]. Chen et al. proposed a congestive force failure (CHF) detection model based on decision tree and support vector machine [25]. Unfortunately, the above work only selected some special values of cardiopulmonary metabolic indicators based on a limited dataset. Furthermore, due to the black-box principle of machine learning, the lack of interpretability seriously affected its wider application in the medical field, in which it is critical for users to understand and trust the logic of the corresponding decision-making [26].

XGBOOST, known in full as extreme Gradient Boosting, shares a fundamental idea with GBDT, which is to grow a tree through constant feature splitting. Compared with the two traditional machine-learning algorithms, SVM and RF, XGBOOST achieves higher accuracy and a higher Macro-F1 Score, and can improve the generalization ability and robustness of a single learner by learning from the residual of the previous tree; thus, it has a better capability of interpretability. Song et al. used XGBOOST to improve a steel property optimization model [27]. Pan used the XGBOOST algorithm to predict hourly PM2.5 concentration [28]. Ogunleye built an XGBOOST model to diagnose chronic kidney disease [29]. The purpose of this study is to build a CPET evaluation model based on XGBOOST and the genetic algorithm (GA), to achieve accurate and explanatory evaluation results based on high-noise and large-volume of data collected from CPET.

## 2. Materials and Methods

### 2.1. Participants

A total of 120 young men (21 ± 3.5 yrs.) with good physical fitness were selected for this experiment. The healthy male subjects were within the height range of 174 ± 3.51 cm, weight range of 65.2 ± 8.44 and BMI range of 21.55 ± 2.90 kg/m^2^. Before the experiment, physical screenings were conducted on the subjects, to ensure that there were no signs of the following diseases: cardiovascular disease; bone disease and metabolic system disease. All subjects were instructed to avoid alcohol, to exercise vigorously, and to obtain sufficient sleep for the first 24 h of the experiment. All subjects signed the consent and notified about the potential risks in the experiment. 

### 2.2. The Cardiopulmonary Test (CPET)

The CPET test was conducted on the Quark CPET measurement platform (COSMED Co., Rome, Italy). The entire exercise process for all participants was completed under the supervision of professional medical staff in the hospital to guarantee reliable and highly comparable data. All subjects in this experiment completed the maximum exercise test on a power bicycle, as shown in Figure 1. As it is difficult to maintain a constant bicycle speed throughout the test, the bicycle speed is a component of power-per-minute, subjects were required to maintain a maximum of 55 to 65 r/min during the pedaling. After a 3 min rest period and a 3 min warm-up period, subjects pedaled at a rate of 30 w/min until exhausted.

Before the CPET data were used to train the automatic assessment model, a classification standard was set up, as Figure 2 shows. A CPE evaluation index system was first established and then optimized. The weights to the indexes were assigned based on the C-OWA operator and entropy weight method, as shown in Table 1 [30]. The TOPSIS-RSR method was used to determine subjects’ CPE rating levels [30]. Finally, GA-XGBOOST was used to optimize the weight matrix.

### 2.3. Algorithms

In this study, XGBOOST was used as the classification value of the model: input training set X, namely CPET data and output tag set. The first and second derivatives of each sample were calculated. The tree with the maximum Gain was selected by greedy strategy, and the sum of all the first and second derivatives of the tree can be used to calculate the classification value of the sample. Its process was approximately shown in Figure 3. The hyperparameter settings of the XGBOOST algorithm were as follows: the default base learner was GB-tree; the number of base learners was 100; the learning rate is 0.1; and the other parameters were default values. Detailed code for XGBOOST and genetic algorithms is provided in the Appendix A.

The genetic algorithm was used to optimize the machine learning model. The steps of the genetic algorithm are shown in Figure 4. The parameters of XGBOOST were categorized according to 3 types: general parameters; BOOST parameters; and learning target parameters. Among the BOOST parameters, the number of iterators (N_estimator), the learning rate (Learning_rate) and the maximum depth of the number (max_depth) largely determine the performance of the model; thus, they were optimized in modeling. After setting the genetic algorithm parameters, the genetic operator was determined and the Macro-F1 Score was selected as the fitness function, with the maximum value as the optimal solution of the genetic algorithm.

The RF algorithm mainly optimizes the number of the decision tree and the maximum feature ratio. The number 100 was selected for this decision tree, and the root of the maximum feature ratio was selected for the number of features. Other parameters were given default values. In the SVM algorithm the penalty function was 1; the kernel function was linear kernel function; the kernel function constant was 0; the highest term coefficient was 0; and the multiclassification strategy was “1 pair of complementary classes”.

### 2.4. Performance Analysis

#### 2.4.1. Tenfold Validation

The most popular algorithms—support vector machine (SVM), random forest (RF) and extreme gradient enhancement (XGBOOST)—were trained and validated the performance. A total of 120 subjects were divided into the categories of training set and test set in a ratio of 7:3. To ensure that the final evaluation model was not affected by this process, the machine learning model was constructed and screened by using ten-fold cross validation (rounded if it is not an integer) in the training set. The model with the best performance will be assessed by the test set. All the calculations were run on Python 3.6.2 (https://www.lfd.uci.edu/~gohlke/pythonlibs/#xgboost (accessed on 10 June 2022)).

#### 2.4.2. Statistical Analyses

The Macro-Average method was used to evaluate the performance of the machine learning model. Accuracy, Macro-Precision, Macro-Recall and Macro-F1 Score were applied to evaluate the model performance:(1)$$\mathrm{Accuracy}=\frac{TP+TN}{TP+TN+FP+FN}$$
(2)$$\mathrm{Macro}\;\mathrm{Precision}=\frac{1}{n}\sum_{i=1}^{n}\mathrm{Pi}$$
(3)$$\mathrm{Macro}\;\mathrm{Recall}=\frac{1}{n}\sum_{i=1}^{n}Ri$$
(4)$$\mathrm{Macro}\;F1=\frac{1}{n}\sum_{i=1}^{n}F1i$$

#### 2.4.3. Confusion Matrix

The test set was used to evaluate the results of the XGBOOST model classification. Confusion matrix is a commonly used auxiliary tool in machine learning, which can intuitively understand the prediction of each category in the sample. The darker the color, the greater the density/number and the better the prediction accuracy.

## 3. Results

### 3.1. Performance of Classifying Models

As Table 2 shows, XGBOOST has the highest accuracy rate. The three machine learning models can identify the CPE level of the human body, and the scores of the Macro-F1 are all above 81%. Among them, the XGBOOST algorithm has achieved better results than the first two algorithms in both accuracy and the Macro-F1 Score.

The confusion matrix of XGBOOST to test set classification is shown in Figure 5. Figure 5a shows the confusion matrix predicted by XG-BOOST. The actual number of subjects with average, good and excellent levels was 12, 13 and 11, respectively, while the predicted number of subjects with those three levels were 11, 14 and 9, respectively. This means that the model may be more sensitive to the judgment of intermediate levels. Figure 5b showed the process in more detail: subjects 2, 19 and 35 are overestimated, while subjects 29 and 34 are overestimated. These may be due to individual errors and personal factors. The accuracy of test set recognition was 0.861 and the Macro-F1 Score was 0.864. XGBOOST correctly identified all excellent individuals but failed to correctly identify three “good” and “general” individuals.

### 3.2. Hyperparameter Optimization of XGBOOST Based on Genetic Algorithm

The official documentation for XGBOOST divides the parameters of XGBOOST into three categories: general parameters; BOOST parameters; and learning target parameters. Among the BOOST parameters, the number of iterators (n_estimator), the learning rate (learning_rate) and the maximum depth of the number (max_depth) largely determine the performance of the model. In this study, the genetic algorithm was used to optimize the three variables of XGBOOST, with the following steps:Step 1: Set genetic algorithm parameters

The individuals were coded in binary. The initial population and maximum iteration times were 50 and 100, respectively. The crossover probability PC was set as 0.95, and the mutation probability PM was set as 0.05. The gene length was 13;

Step 2: Determine genetic operators

Crossover operators: Single-point crossover was selected for the first three and four–six loci, and multi-point crossover was selected for the last six loci. Suppose that four crossover points are randomly generated (point 2, point 4, point 9 and point 12), two new individuals can be generated by crossing the two, as shown in Figure 6.

Mutation operator: Single-point variation was selected for the first three positions and positions four–six of the mutation operator, and uniform variation was selected for the last six positions. Suppose that four crossover points (point 2, point 4, point 9 and point 12) are randomly generated, then the individual generated after the mutation of the altered individual is shown in Figure 7.

Selection operator: To avoid the elimination of excellent individuals, this study adopted the ranking method and elite strategy to “recommend” the top 5% individuals to ensure that they were not eliminated;

Step 3: Determine fitness function

The Macro-F1 Score was selected as the fitness function, and the maximum value was used as the optimal solution for the genetic algorithm. The genetic algorithm was used to optimize the XGBOOST algorithm, and the fitness iteration curve is shown in Figure 8. The best Macro-F1 Score was 0.915. Table 3 lists the optimized parameters.

Using the genetic algorithm to adjust parameters for three times, the results of the 50th iteration are better than the default parameters of XGBOOST, as shown in Figure 8. The accuracy and the Macro-F1 Score of XGBOOST, optimized based on the genetic algorithm, are improved by 6.5% and 5.9%, respectively.

### 3.3. Interpretability of GA-XGBOOST Output

SHAP is applied to explain the importance of features in GA-XGBOOST established in this paper. Shapley value can describe the importance of each feature when the model makes a prediction on specific samples. Compared with traditional feature-importance description methods, SHAP has better consistency [31]. The SHAP value of each feature is drawn for each sample, as shown in Figure 9. The ordinate signifies different physiological characteristics, the abscissa signifies the SHAP value, a point represents a sample, and the color represents the feature value (red is high, blue is low). The absolute average SHAP value of each feature is taken as the importance of the feature, as shown in Figure 10.

The results show that the first five feature parameters—$X_{12}$, $X_{6}$, $X_{7}$, $X_{2}$,$\;\mathrm{and}\;X_{16}$; namely, VO2_MAX, OP_MAX, VE/VO2_AT, HR_MAX, and C(a-v)O2_MAX—have the greatest impact on the classification results. Among them, maximal oxygen uptake has the greatest influence on the model. The higher the characteristic value, the lower the SHAP value, which can improve the CPE of the subjects; when using the XGBOOST model for classification, “1” is excellent, “2” is good, and “3” is general, as mentioned above. $X_{6}$, the oxygen pulse at exhaustion, is the second important characteristic, and the higher the value of this characteristic, the greater the CPE. $X_{9}$ signifies the ventilation at the anaerobic threshold, which is the least important feature. The above experimental results are consistent with the clinical and related findings in the previous review [5,6,10,11,12]. Figure 11 shows the SHAP feature-dependence diagram of the two indicators. (VO2_MAX) is approximately positively correlated with the target variable (C(A-V)O2_MAX). As shown in Figure 11b, (VE/VO2_AT) is negatively correlated with the target variable (VO2_MAX). Baba R pointed out that any subject with a defective motor system would decrease VO2, which leads to a rapid increase in oxygen equivalent [29]. This is consistent with the results explored in this study.

## 4. Discussion

The goal of the current study is to develop and validate a computer-aided algorithm for automatic evaluation of CPET test results and to obtain a relatively reasonable and accurate model. In this study, it is shown that a wide range of 16 CPET parameters, after being automatically selected and weighted, are fed into the model, generating quite an accurate CPE level assessment, to distinguish between the “Excellent”, “Good” and “General” individual CPE level. Different classification scaling methods have been proposed to distinguish CPE in different populations: Guo et al. studied the CPE of adults and divided them into the categories of excellent, good, medium, poor and very poor [32]. ACSM’s Health-Related Physical Fitness Assessment Manual classifies adult males by percentile: excellent (95–99%); very good (80–95%); good (60–80%); general (40–60%); weak (20–40%); and quite weak (0–20%) [32]. Considering the general requirements for high-latitude on-site work, the three level granularity is sufficient [33].

A series of data (WR, VO2/kg, HR, and VE, etc.) provided by CPET may obtain different results in different expert diagnoses, which means that manual interpretation can be very confusing and subjective. As such, a machine learning approach on analysis of all CPET data can diminish the differences between individuals’ expertise and achieve a relatively constant criteria. The results showed that the CPE evaluations based on XGBOOST were accurate and consistent with the clinical diagnosis results of expert scoring. Distinct from the subjective selection of CPET parameters in other studies, this study determines the physiological parameters of the selected subjects through Pearson correlation analysis, principal component analysis (PCA), the TOPSIS-RSR rating evaluation and the C-OWA weighting method. Furthermore, it generates a classification scale for the assessment of the CPE rating. The complete process proves the feasibility of fully automatic CPE assessment. CPET possesses a major advantage over traditional cardiopulmonary endurance assessment in that it provides multiple parameters related to cardiopulmonary endurance, which facilitate a more comprehensive assessment rather than relying on a single parameter. In view of the weaknesses of CPET (its large amount of data and strong subjectivity in expert evaluation), the use of a machine learning algorithm for modeling can well solve these two problems and achieve the goals of high accuracy, speed and objectivity.

At present, many CPET-related studies have been published; however, only a few have been successfully modeled using machine learning methods. Traditional CPET research methods mainly use statistical methods to generate a general result from multivariate CPET and cannot be traced back for an explanation on how key features are connected to the assessed classification level. In Ong’s research, CPET data were generalized to indicate the serious level of chronic obstructive pulmonary disease (COPD) but could not indicate which parameters doctors should mainly focus on to treat the syndrome of this disease [34]. Several research studies have used machine learning to investigate CPET and have yielded considerable results: A sequential clustering method proposed by Li et al. selected CPET data including oxygen consumption, carbon dioxide, heart rate and stroke output for training, but the actual recognition effect of the model was not good. Deak et al. proposed a fuzzy evaluation method based on multiple physiological parameters such as VO2max, SV and HR, but it could not quantitatively evaluate the CPE level of the body. The contribution of this study is that relatively high accuracy was achieved after the qualitative classification method was proposed; unlike the traditional regression algorithms currently available, the neural network model used in this study can constantly learn from new data, and its predictive power can be improved by providing new CPET experimental data. In our study, the correlation analysis method was used to screen and analyze the importance of each CPET parameter, and then a model was established based on XGBOOST, which proved to be relatively successful in the verification stage. As in many scientific tasks and other complex tasks correct models will undoubtedly help facilitate medical diagnosis and implement appropriate measures that are less susceptible to the subjective judgment of experts or doctors. Current efforts represent a practical possibility for medical interpretation software to aid in objective assessment.

Some study limitations are to be noted. First, a comparatively small sample size may increase the error, and the model accuracy may also be affected by the quality of CPET original data. The accuracy and the Macro-F1 Score of the XGBoost CPE assessment model in this study were not particularly ideal. Therefore, more experimental data can be collected in the future to modify the model: on the one hand, to study how to select physiological characteristic parameters, and on the other hand, to optimize the parameters of XGBOOST. Secondly, the measurement error caused by the technical limitation of experimental equipment should not be ignored. To fit the model into the personnel placement and management of high-latitude worksites, future work should be compared with more CPE testing protocols and high-latitude work factors should be added into the protocols to train and test the model. Finally, although machine learning has achieved good success in many fields, its wider application in some security-sensitive application fields is affected by the lack of interpretability, due to the black-box principle of machine learning. In high-latitude on-site workers’ selection, training and occupational prevention, it is of vital importance to build up a model with good interpretability. Based on the advantage of XGBoost’s capacity to interpret the black-box output, this study has proved the possibility of automatic interpretation from various information granularity. Further studies can be completed to train the model on further diagnostic expertise.

## Figures and Tables

**Figure 1 diagnostics-12-02538-f001:**
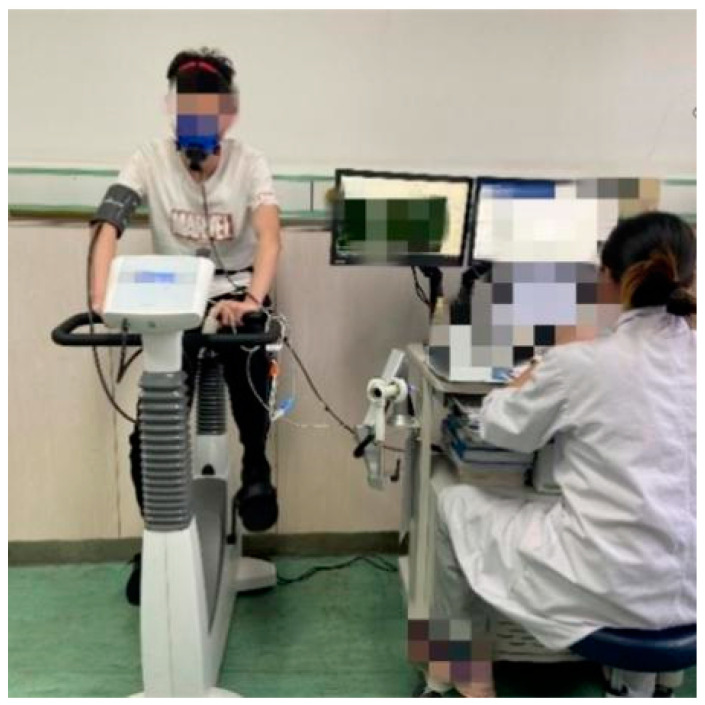
Illustration of cardiopulmonary exercise testing.

**Figure 2 diagnostics-12-02538-f002:**
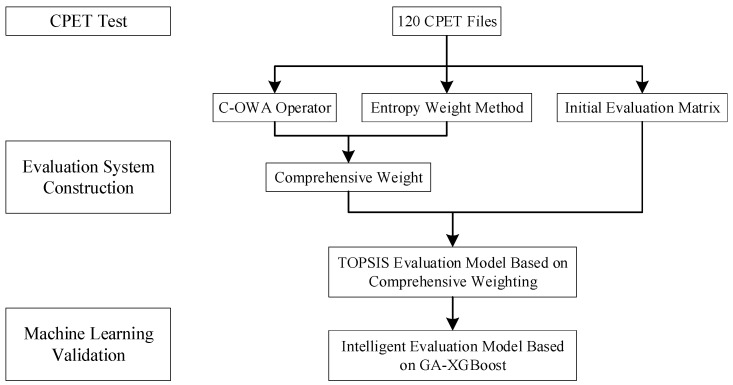
Schema of the study’s design.

**Figure 3 diagnostics-12-02538-f003:**
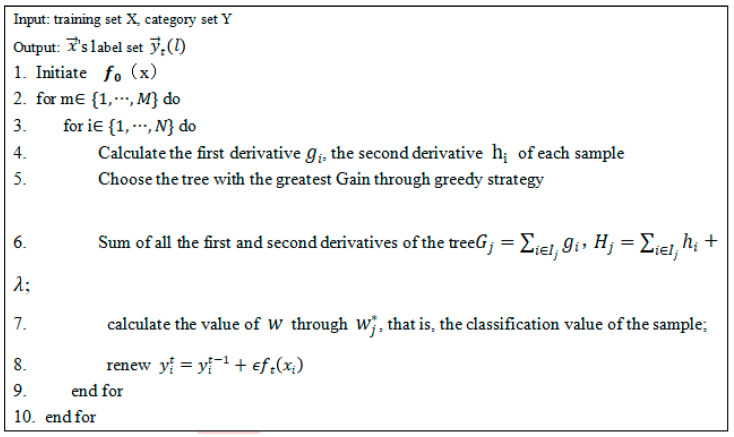
Pseudocodes of the XGBOOST algorithm.

**Figure 4 diagnostics-12-02538-f004:**
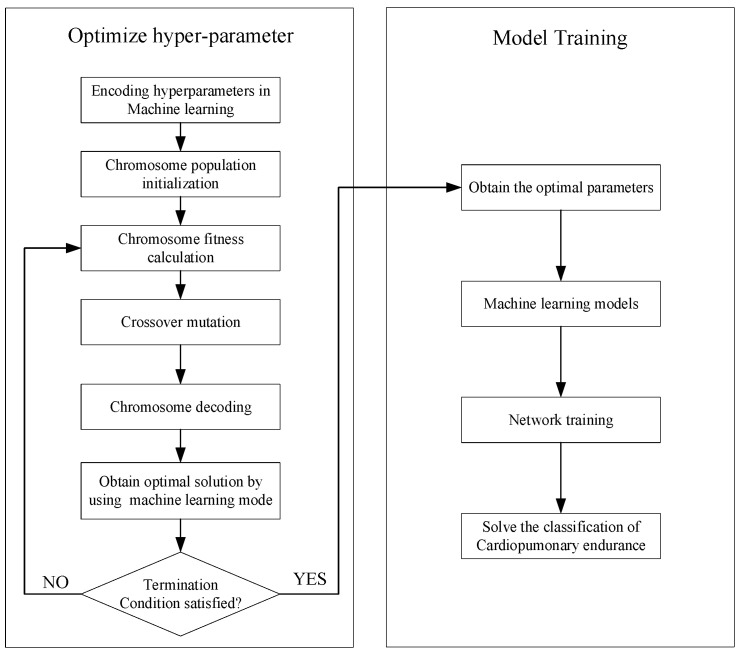
Optimization Flow Chart of Genetic Algorithm.

**Figure 5 diagnostics-12-02538-f005:**
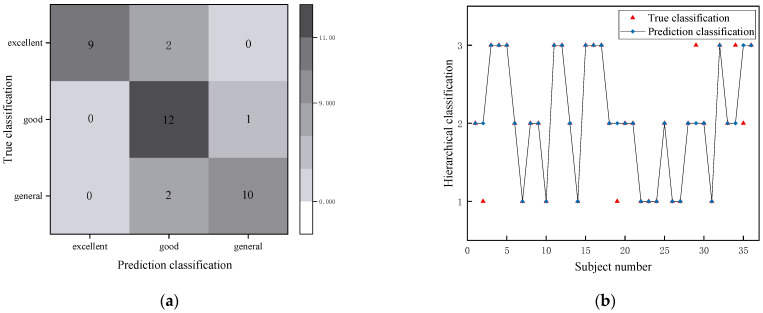
(**a**) Confusion matrix predicted by XGBOOST; (**b**) Hierarchical classification.

**Figure 6 diagnostics-12-02538-f006:**
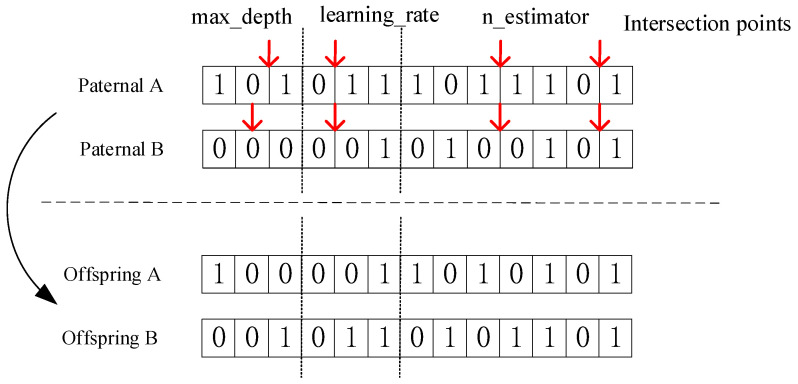
Crossover Operator.

**Figure 7 diagnostics-12-02538-f007:**
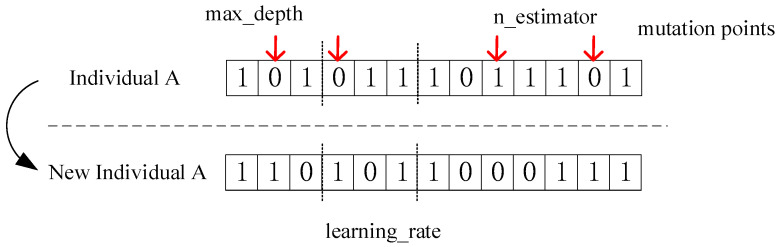
Mutation Operator.

**Figure 8 diagnostics-12-02538-f008:**
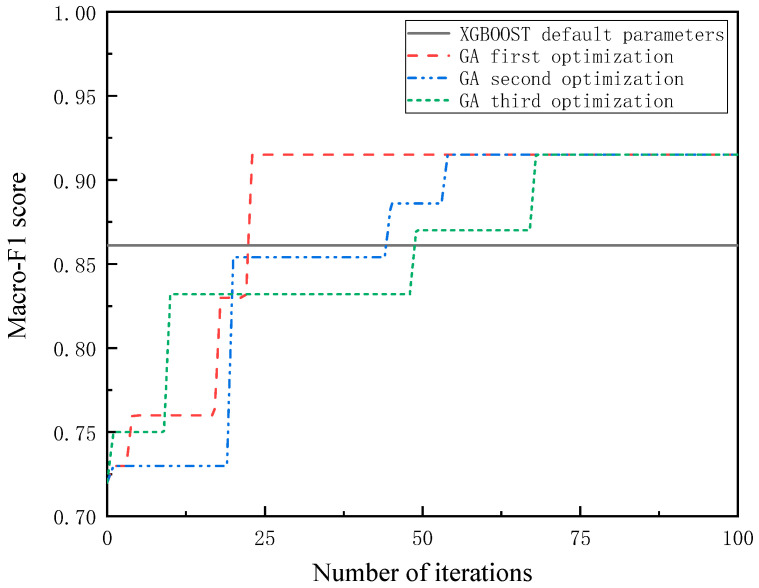
Iteration diagram of genetic algorithm.

**Figure 9 diagnostics-12-02538-f009:**
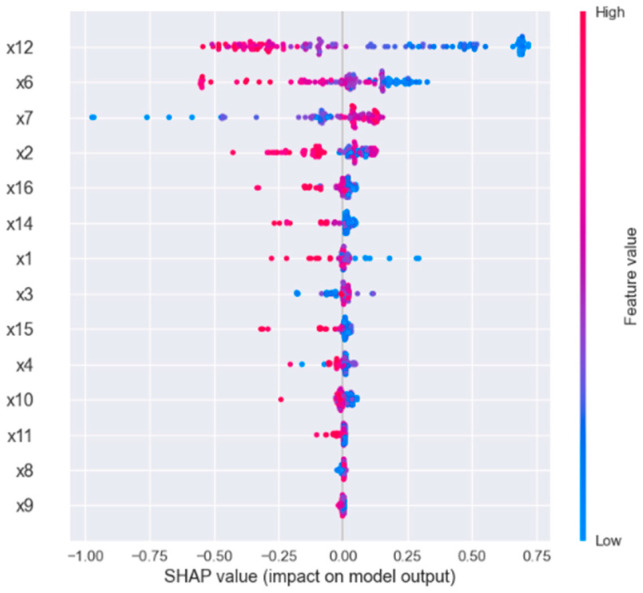
SHAP value of each feature for each sample.

**Figure 10 diagnostics-12-02538-f010:**
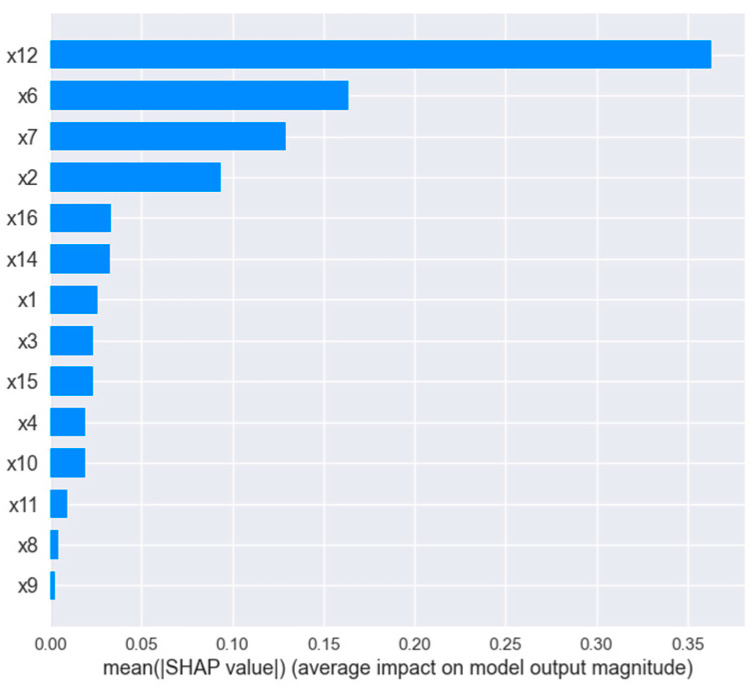
Importance of each feature.

**Figure 11 diagnostics-12-02538-f011:**
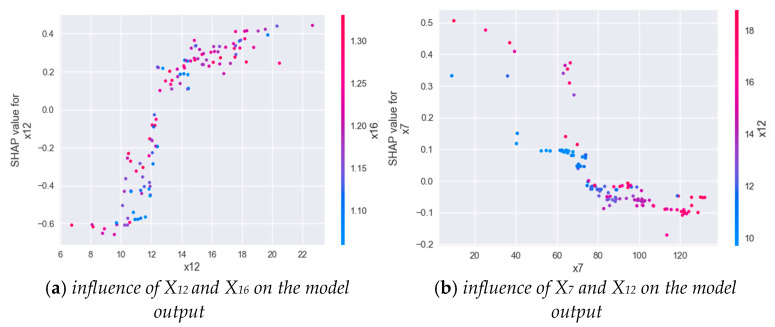
SHAP feature dependency graph examples of important indicators.

**Table 1 diagnostics-12-02538-t001:** Determination of the weights of subjective and objective indicators.

	Index	Weights
Cardiovascular Capacity	$\mathrm{HR}\_\mathrm{AT}(X_{1}$)	0.078
	$\mathrm{HR}\_\mathrm{MAX}\left(X_{2}\right)$	0.058
	$\mathrm{SV}\_\mathrm{AT}\left(X_{3}\right)$	0.073
	$\mathrm{SV}\_\mathrm{MAX}\left(X_{4}\right)$	0.048
	$\mathrm{OP}\_\mathrm{MAX}\left(X_{6}\right)$	0.075
Respiratory Metabolic Capacity	$\mathrm{VE}/VO_{2}\_\mathrm{AT}\left(X_{7}\right)$	0.096
	$\mathrm{VE}/VO_{2}\_\mathrm{MAX}\left(X_{8}\right)$	0.064
	$\mathrm{VE}\_\mathrm{AT}\left(X_{9}\right)$	0.075
	$\mathrm{VE}\_\mathrm{MAX}\left(X_{10}\right)$	0.087
Metabolic Capacity	$VO_{2}\_\mathrm{AT}\left(X_{11}\right)$	0.064
	$VO_{2}\_\mathrm{MAX}\left(X_{12}\right)$	0.111
	$\mathrm{RQ}\_\mathrm{MAX}\left(X_{14}\right)$	0.053
	$C\left(a-v\right)O_{2}\_\mathrm{AT}\left(X_{15}\right)$	0.054
	$C\left(a-v\right)O_{2}\_\mathrm{MAX}\left(X_{16}\right)$	0.066

**Table 2 diagnostics-12-02538-t002:** Performance parameters of machine learning algorithms.

Model	Accuracy	Macro-Recall	Macro-Precision	Macro-F1 Score
RF	0.810	0.810	0.886	0.812
SVM	0.843	0.843	0.876	0.841
XGBOOST	0.861	0.861	0.879	0.864

**Table 3 diagnostics-12-02538-t003:** Optimal parameters of XGBOOST.

Parameter Name	Meaning	Default Value	Optimal Value
max_depth	The maximum depth of number	10	221
learning_rate	Learning rate	0.1	0.285
n_estimator	Number of iterators	100	3

## Data Availability

The data used to support the findings of this study cannot be made freely available to protect patient privacy. Requests for access to these data should be made to the corresponding author.

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
