# Peer review of "Automatic Cardiopulmonary Endurance Assessment: A Machine Learning Approach Based on GA-XGBOOST"

_diagnostics, 2022, doi:10.3390/diagnostics12102538_

Round 1

Reviewer 1 Report

The authors aimed to build a CPET evaluation model based on XGBOOST to tackle the high noise of data collection of larger volumes. The current study may spark interest from the readers. However, I have major comments to address. 

My major concern is regarding the limitations of the machine learning approach to evaluate the CEPT dynamics. As relevant as the maximal (or peak) values cardiopulmonary kinetics have applications in sports and clinical fields. Is it considered in your approach?   

Introduction: It is too long and difficult to follow. Please provide a clear rationale highlighting the gaps in the literature and the study's originality.

Lines 34-35: The stress induced by intravenous pharmacological stimulation has different applications compared to CPET. These two different tests should be not put together as done by the authors.

Lines 43-45: There are several tests with or without exercise models to predict the maximal aerobic capacity (VO2max) in several populations. Thus, these tests do not depend on medical professional assistance. Thus, is a prediction by machine learning approach necessary for this field?

Materials and methods

Lines 129-131: Did all participants pedal at a fixed rate of 30w/min until exhaustion? Why an individualized protocol is not employed?  

Table 1: Why did the authors use maximal values rather than peak values?

Discussion: This section should go more depth into the feasibility of the proposed model for clinical and sports applications. The authors discuss why an "'automatic" cardiopulmonary endurance assessment could be superior to traditional methods (e.g., considering CPET and predictive VO2max models).

Author Response

1. My major concern is regarding the limitations of the machine learning approach to evaluate the CEPT dynamics. As relevant as the maximal (or peak) values cardiopulmonary kinetics have applications in sports and clinical fields. Is it considered in your approach?    

Answer: The main focus of our experiment is CPET dynamics. As for aspects of cardiopulmonary kinetics, this is a very interesting research direction worth exploring. We will consider this in future research. Thanks for your suggestions.

2. Introduction: It is too long and difficult to follow. Please provide a clear rationale highlighting the gaps in the literature and the study's originality.

Answer: We modified the introduction part in case that the gaps are clearly highlighted. As the text shows, the rationale for this study is stated from 2 aspects: to analyze how CPET data is inefficiently analyzed, and to analyze the challenged for AI methods to process CPET data for automatic CPET diagnosis.

3. Lines 34-35: The stress induced by intravenous pharmacological stimulation has different applications compared to CPET. These two different tests should be not put together as done by the authors.

Answer: We agree that the two are quite different in data collection paradigm and focus on different aspects of cardiopulmonary capability, which should be reflected by different features. It cannot provide comparable hints for our research protocol. We remove the part of intravenous pharmacological stimulation.

4. Lines 43-45: There are several tests with or without exercise models to predict the maximal aerobic capacity (VO2max) in several populations. Thus, these tests do not depend on medical professional assistance. Thus, is a prediction by machine learning approach necessary for this field?

Answer: VO2max is a very important parameter, but it is only one of the physical parameters provided by CPET. There are other parameters (e.g., Anaerobic Threshold) that are also important for cardiopulmonary endurance assessment. Thus, we need to use machine learning to make more rapid, efficient, accurate and objective predictions based on the rich, redundant data collected on CPET exercise models. For more reasonings, please to the introduction part, where the significance of CPET is stated in details.

Materials and methods

5. Lines 129-131: Did all participants pedal at a fixed rate of 30w/min until exhaustion? Why is an individualized protocol not employed? 

Answer: We used subjects with similar physical conditions and one of our purposes of this study is to build up a standard job requirement for workers in high-latitude places. After carefully consulting with the medical and task experts, a fixed rate of 30W /min until exhaustion protocol is designed to collect a standardized database, reducing the effect of the individual variation.

6. Table 1: Why did the authors use maximal values rather than peak values?

Answer: Sorry about the confusion. The maximal and peak here both means the highest value. We make changes and replace all with maximal values.

7. Discussion: This section should go more depth into the feasibility of the proposed model for clinical and sports applications. The authors discuss why an "'automatic" cardiopulmonary endurance assessment could be superior to traditional methods (e.g., considering CPET and predictive VO2max models).

Answer: We add the following into the discussion part to restate the devotion of our study: A major advantage of CPET over traditional cardiopulmonary endurance assessment is that it provides multiple parameters related to cardiopulmonary endurance, which facilitates a more comprehensive assessment rather than relying on a single parameter. In view of the weaknesses of CPET (large amount of data and strong subjectivity in expert evaluation), we use machine learning algorithm for modeling, which can well solve these two problems and achieve the goals of high accuracy, speed and objectivity.

Reviewer 2 Report

1. In introduction or method, please describe GA-XGBOOST and how differences between GA-XGBOOST and regular XGBOOST

2. GA-XGBOOST stand for?

3. The authors have not described the software that they used, python vs R? What packages, need to describe in details with citations.

4. Traditional statistical models, multivariate, should also be used as comparison to these models.

5. ROC curves should also be provided

6. calibration curves should also be provided 

7. Models should be provided or applied to shinyapps or supplementary codes for transparency

Author Response

1. In introduction or method, please describe GA-XGBOOST and how differences between GA-XGBOOST and regular XGBOOST

Answer: As our preliminary results show, although XGBOOST performed out of the two other algorithm, it still has a space to be optimized. Often the hyperparametric algorithms can be optimized by grid search, random search, heuristics, etc. Compared with grid search and random search, heuristic algorithm takes less time and training times. Among heuristic algorithms, compared with simulated annealing algorithm and ant colony algorithm, genetic algorithm can achieve more than 90% of the optimal solution at a very fast speed. Therefore, genetic algorithm is used to optimize the XGBOOST model in this study.

2. GA-XGBOOST stand for?

Answer: GA-XGBOOST is to indicate optimize the parameters of XGBOOST using genetic Algorithm. We add the full name in the text.

3. The authors have not described the software that they used, python vs R? What packages, need to describe in details with citations.

Answer: We used python. The packages can be found on https://www.lfd.uci.edu/~gohlke/pythonlibs/#xgboost .

4. Traditional statistical models, multivariate, should also be used as comparison to these models.

Answer: We added the following text into discussion to clarify how our method is different from traditional models: Traditional CPET research methods mainly use statistical methods to generate a general result from multivariate CPET and cannot be traced back to show what variables are the key features deciding the classification level. In Ong’s research, CPET data were generalized to indicate the seriousness level of chronic obstructive pulmonary disease (COPD) but could not indicate what parameters the doctors should be mainly focused to treat the syndrome of this disease [34].

5. ROC curves should also be provided

Answer: As our study is a multi-class classification problem. We would rather use our own way to presenting the comparison between the good and bad, as shown in Fig5 (b)

6. calibration curves should also be provided 

Answer: We used two different ways of calibration in Fig. 5(a) and Fig. 5(b) to reflect how the predicted result deviated from the actual result. believe that the two graphs in Figure 5 can well replace the functions of ROC curve and calibration curve.

7. Models should be provided or applied to shiny apps or supplementary codes for transparency

Answer: We will attach the relevant code as the appendix of the text: xgboost_CPET.py and xgboost_shap.py.

Round 2

Reviewer 1 Report

My all concerns were addressed by the authors.